# Population Status and Vulnerability of *Mantidactylus pauliani* from Ankaratra Protected Area, Madagascar

**DOI:** 10.3390/ani13172706

**Published:** 2023-08-25

**Authors:** Herizo Oninjatovo Radonirina, Bernard Randriamahatantsoa, Nirhy H. C. Rabibisoa

**Affiliations:** 1Doctoral School of Natural Ecosystem, University of Mahajanga, Mahajanga 401, Madagascar; 2Environmental and Life Science, Faculty of Sciences Technology and Environment, University of Mahajanga, University Campus of Ambondrona, Mahajanga 401, Madagascar; bernardzoo01@gmail.com

**Keywords:** *Mantidactylus pauliani*, Ankaratra, landscape, habitat type, season, distribution

## Abstract

**Simple Summary:**

*Mantidactylus pauliani* is a locally endemic amphibian species restricted to mountain streams on the Ankaratra Massif, in the central highlands of Madagascar. This species has a highly restricted distribution, which makes it vulnerable to habitat destruction and, consequently, *M. pauliani* is considered one of the most threatened frog species in Madagascar. Therefore, having information concerning *M. pauliani* and its habitat is necessary for effective conservation. Our study aimed to examine the population status of *M. pauliani* by verifying its geographic distribution and elevational range, exploring habitat use, and assessing threats. *M. pauliani* occurs from 1900 m to 2378 m a.s.l. and is most abundant at an altitude between 1993 and 2166 m. Adults, juveniles, and tadpoles were associated with different levels of stream depth, speed, and width. We found that human activities are contributing to habitat losses, which is modifying its environment and threatening the species’ survival. The data collected on the occurrence and habitat use of *M. pauliani* serve as an “environmental alert dashboard” to promote sustainable conservation.

**Abstract:**

Mountain summits in Madagascar generally have species with specific habitat requirements, providing a home to a unique and locally endemic herpetofauna. Among them is *M. pauliani*, a typically aquatic and critically endangered amphibian found on the Ankaratra Massif. This species inhabits high elevations with a limited distribution range. Our study aimed to present new data on the distribution and elevational range, habitat use, and threats to *M. pauliani* and its occurrence according to habitat changes. To achieve this, annual monitoring was carried out from 2018 to 2021. Nine 100 m transects were established along streams at elevations ranging from 1762 to 2378 m a.s.l. along which we conducted visual encounter surveys. Data analysis was performed using a χ^2^ test and Factor Correspondence Analysis. We found that *M. pauliani* occupies elevations between 1900 and 2378 m a.s.l. within humid forests and savannah habitats. The results showed a fluctuation in the number of animals observed and a higher occurrence at higher elevations throughout the years according to the season, stream quality, and water volume. Ongoing habitat alteration makes *M. pauliani* vulnerable to population decline, with annual bushfires likely having a negative impact on habitat.

## 1. Introduction

The Ankaratra forest, one of the biodiversity hotspots in Madagascar, is home to the locally endemic and Critically Endangered amphibian, *Mantidactylus pauliani* [1,2]. This species has previously been recorded at elevations above 2000 m [3,4,5], and specifically lives in the cold rocky streams of the Ankaratra Mountain. Since *M. pauliani* is a typically aquatic species [6], its distribution is thus restricted. Hence, it greatly relies on the streams’ availability in terms of quantity and quality for breeding and sustenance [6,7,8].

However, in the last few decades, the forest [9] and its streams have gradually undergone changes due to various human activities, including charcoal production, timber harvesting, cattle grazing, and wildfire [5,10,11]. Anthropogenic activities increase threats to the species’ habitats, leading to disruption in population connectivity [12], alteration, and degradation [9], which may induce vulnerability as amphibians are among the most highly threatened species in the world [13].

Considering the limited distribution of *M. pauliani* and its highly dependent behavior on streams, it is likely to be sensitive to habitat change [14], making it an important indicator of environmental health. However, the impact of habitat change on this locally endemic frog remains unknown because the data are still insufficient [12]. Long-term studies are also needed to produce reliable information for helping conservation activities to mitigate the vulnerability from threats that have a negative impact on the *M. pauliani* population.

The aim of this study was to present the new data on the spatial distribution and occurrence, habitat use, and threats to *M. pauliani* within a spatiotemporal approach. Based on previous research [5,12,14] and the theoretical considerations, we hypothesized that habitat degradation can change the distribution and occurrence of the *M. pauliani* population. More specifically, this study explored the relationship between this species and their occurrence by focusing on their spatio-altitudinal distribution and ecological parameters, once the knowledge on habitat use and threats has been identified. The new information gathered during this research can help us understand the ecological guilds and abundance variation of this critically endangered frog. While *M. pauliani* reproduces during both humid and dry seasons, the humid season represents the seasonality in months, and the variability is significant. These data are needed to produce an effective and sustainable conservation action plan.

## 2. Materials and Methods 

### 2.1. Study Area 

The study was conducted within the New Protected Area (NPA) of Ankaratra, which is one of the few remaining forests in the central highlands of Madagascar, covering about 8130 hectares (Figure 1). It extends over the slopes of the Ankaratra Massif, situated between 19°19′ and 19°24′ south latitude and 47°14′ and 47°22′ east longitude. The study area is part of the Manjakatompo Forest Station, located 84 km from the city of Antananarivo, via the RN7, and 17 km west of the city of Ambatolampy.

### 2.2. Data Collection

We conducted four 15-day field trips in September 2018, March 2019, February 2020, and July 2021. These four trips made it possible to monitor variations in the population size of *M. pauliani* according to the season (dry and humid) and the different habitats. The different types of ecological guilds, the presence of permanent rock formations in the streams, and elevation levels were taken into account when selecting the study sites. The sites cover the entire mountain range (crest, valley, and slope) and differ in terms of habitat characteristics and degree of degradation. Six sites were selected in 2018 and 2019, and three additional sites were included in 2020 and 2021, namely Maharavana, Analafohy, and Ambohimirandrana (Table 1).

A 100 m transect was established at each site with varied widths based on the structure of the stream. We gathered data along the transects through direct observation and by examining shelters and refuges [15]. A transect consisted of a 100 m line that ran along the stream and was marked every 10 m with a flag. By surveying along transects, we were able to count individuals and categorize them by life stage and sex.

To define the stream parameters, water depth and speed were measured using the same techniques as reference [16]. To calculate the speed, a floating object (typically a cork) was timed over a one-meter distance. A straight stick held vertically was used to measure the distance between the surface and the bottom of a stream at a stable spot with buried rock. Three samples were gathered where the current speed was timed using the benchmark “flags”. During each field trip, the same parameters were recorded at the same locations.

### 2.3. Data Analysis 

Data analysis was performed using Microsoft Excel. A Chi-square test (χ^2^) was used to confirm whether the elevational distribution of *M. pauliani* for each developmental stage was significantly different according to the study season. It was also performed to determine if the numbers of this species by developmental stage vary by season. In addition, a multivariate analysis was carried out using correspondence factor analysis, which is a method for analyzing data when the variables to be studied are quantitative measures. We used correspondence factor analysis to evaluate the habitat use of the individuals for each developmental stage and their characteristics according to the physical parameters of the streams. It consisted of projecting the individuals (adults, juveniles, and tadpoles) and the variables, such as water speed ranges (0.5–1 m/s, and 1–1.5 m/s, water depth ranges (0–20 cm, 20–40 cm and 40–60 cm), stream width (0–1 m, 1–2 m and 2–3 m), slopes (low slope, medium slope, and steep slope), microhabitat (still water, bank, and rock), and stream substrates (muddy, rocky, and sandy), which can influence the distribution of *M. pauliani* on a factorial plan. 

We identified the threats to the species’ habitat using a model adapted from reference [17]. The model was used to determine the degree of each threat by assigning a score of 1–3 points for each threat at each study site. The variables scored included their duration (permanent with a score of 3 points, temporary with 2 points, or occasional with 1 point), their intensity (strong, medium, or weak), and their importance (regional, zonal, or local). The assessment was based on the sum of these degrees of threat (Dm) according to duration, intensity, and the importance for each type of disturbance. If Dm ≤ 4, the pressure is minor; if 4 < Dm < 7, the pressure is medium; and if Dm ≥ 7 the pressure is major. Then, the pressure index (PI) was calculated to determine the overall importance of each type of disturbance. The PI is between the total number of points assigned to a site and the maximum value of the site [18]. The value of the index varies from 0 to 1. If 0.8 < PI ≤ 1, the threat is very high; 0.7 < PI ≤ 0.8, the threat is high; 0.5 < PI ≤ 0.7, the threat is medium; 0.3 < PI ≤ 0.5, the threat is low; and 0 < PI ≤ 0.3, the pressure is very low.

## 3. Results

### 3.1. Spatial Distribution and Elevational Range 

Table 2 shows the total numbers of individuals of *M. pauliani* recorded over four years (2018, 2019, 2020, and 2021). For all developmental stages, the number of individuals encountered varied significantly by season (χ^2^ = 462.89, *p* < 0.0001, df = 6). Adults were more abundant than juveniles and tadpoles in the dry seasons (2018, 2021), while tadpoles outnumbered adults and juveniles in the humid seasons (2019, 2020). During the dry season (2021), more juveniles than tadpoles were observed (101 vs. 48).

In terms of altitudinal distribution (Table 2), *M. pauliani* were recorded at an altitude between 1900 m and 2373 m. However, we observed most individuals at an altitude between 1993 and 2200 m, and the species was particularly numerous at 1993 m altitude during the years 2018 and 2019. Depending on season and altitude, the number of individuals encountered gradually decreased over time. This finding was particularly evident at the 2167 m altitude, where 1 adult, 2 juveniles, and 22 tadpoles were counted during the dry season (2018) and only 26 tadpoles during the humid season (2019). No individuals were observed here in 2020 and 2021. Generally speaking, their distribution along the mountain stream varied significantly for all developmental stages depending on the survey seasons (Adult stage: χ^2^ = 222.26, *p*-value < 0.0001, df = 24; Juvenile stage: χ^2^ = 152.52, *p*-value < 0.0001, df = 24; and Tadpole stage: χ^2^ = 382.48, *p*-value < 0.0001, df = 24).

### 3.2. Habitat Use 

For all years and seasons, we observed more individuals along the forest transect than the savannah transect, with the exception of the 2020 humid season, when more tadpoles were observed in the savannah than in the forest (Figure 2). We also noted that the number of individuals decreased during the dry season in 2018 and 2021, both in the forest (adults: 230 vs. 106) and in the savannah (adults: 59 vs. 15). The same result was also observed during the humid season in 2019 and 2020, except for tadpoles, whose numbers were high in 2020 (forest: 86 vs. 135 and savannah: 59 vs. 162; Figure 2). Overall, we observed more males than females in the forest and more females than males in the savannah. However, more males than females were observed in the savannah in the 2021 dry season (9 vs. 6; Table 3).

We found adults, juveniles, and tadpoles to be associated with different habitat variables (Figure 3). In the Factor Correspondence Analysis, the first factorial explained 100% of the total variability, i.e., 98.2% for axis 1 and 1.8% for axis 2. Thus, both factors explain the individual distributions relative to ecological guilds. Adults were more numerous in streams that have a width between 0 and 1 m, a velocity between 0.5 and 1.5 m/s, and a depth between 0 and 20 cm. They were observed on steep slopes close to the riverbank. On the over hand, juveniles were found either under the rocky stream or hidden behind it on a medium slope. The tadpoles were found in deep, still water (40 to 60 cm depth), with a wide watercourse (2 to 3 m), and were observed in areas with sandy substrates.

### 3.3. Vulnerability Analysis 

We observed six threats types, most of which are related to human activities: bush fires, charcoal production, logging, grazing fires, trampling of waterways by cattle, and the expansion of agricultural areas on slopes. These threats lead directly or indirectly to the degradation of the habitat of this species (see Appendix A).

However, the intensity of these different threat pressures varied depending on the type of formation and its proximity to the watercourse. The intensity of the threat was greatest in the open area where the pressure index was high, PI ≥ 0.5 (principally livestock grazing; Table 4). For the forested area, only anthropogenic activities near streams represented a high threat to the species, e.g., the edge for Manotongana, PI = 0.63 (logging and charcoal). Figure 4 demonstrates the relationship between the vulnerability of the species and changes in its habitat caused by anthropogenic pressures. 

According to this diagram and threat analysis (Figure 4), the PI depends greatly on the distance relative to the stream. Indirect threats lead to decreased forest cover, soil denudation, and climate variability. In the long term, this climatic variability leads to changes in microhabitats and species distribution. Conversely, the direct threats cause erosion, which leads to pollution and drying of the streams. 

## 4. Discussion

### 4.1. Spatial Distribution and Elevation Range 

*M. pauliani* is classified as a critically endangered mountain species in decline [19]. The available data suggest a restricted range for this species at an altitude between 2000 and 2300 m [6,11,20], but our study showed that this range extends from 1900 to 2392 m. We offer two hypotheses to explain their extended distributional range. The first is that the search effort was increased because of the addition of a new transect along the slope from 1900 m to 2400 m. The second is that the current study encompassed all seasons (dry and rainy), compared with other studies that have concentrated solely on the breeding period that occurs during the humid season. Although *M. pauliani* was observed at altitudes of 1900 m and above during the study seasons, the greatest numbers of individuals were recorded in the same altitude range as in previous studies. According to reference [21], one of the biological responses to climate change is an upward shift in species distribution due to increased temperatures. Further work should therefore be carried out on this species to examine whether there has been a shift in response to climate change.

There was a difference in the number of individuals at each developmental stage for different periods. During the dry season in 2018 (pre-reproduction period), we observed more adults than juveniles and tadpoles (289 vs. 132, 127 individuals), but there were more tadpoles during the reproduction period (humid season 2019, 2020) than during the other periods (dry season 2018, 2021). This indicates that the juveniles could reach maturity (121 adults vs. 101 juveniles; dry season 2021). It was noted that the number of tadpoles decreased quickly between the humid season (297/2020) and the dry season (48/2021); this might be possibly due to the fast-flowing streams between both periods and their impact on the speed of tadpole metamorphosis [22].

The majority of the recorded individuals were found at the mid-slope (1900 m and 2200 m), where the forests are stable, and where there is a variety of refuges available in the form of rocks, fine sand, and gravel [16]. In addition, we observed more tadpoles in open habitats than in closed habitats. It is possible that strong sunlight could facilitate their hatching and development [23,24]. 

Regarding the habitat preferences of males and females across the survey periods, a higher number of males were found in forested areas at low elevations than in open and savannah habitats at high elevations, confirming the findings of previous studies [25,26]. Therefore, females may attract males to lay their eggs, and it was documented that Mantellinae rarely lay their eggs in water but sometimes just overhanging the water’s surface [3,27].

Regarding seasonal distribution, *M. pauliani* was more abundant during the cold and dry seasons and decreased during the rainy season. This observation is similar to that observed by reference [21]. Furthermore, high-elevation species reproduce throughout the year, unlike species at medium and low elevations, which are inactive during the cold and dry seasons [28]. This explains the abundance of *M. pauliani* throughout the year. However, the authors of reference [29] explained that changes in abundance correlated with short-term and long-term habitat loss. In the case of the Manotongana transect at 2167 m altitude, the habitat was fragmented and degraded due to erosion and the eventual drying of the stream [30]. This is why this site has a very low number of individuals; only one adult male was recorded in September 2018, and no individual has been reported at this site since February 2020. 

### 4.2. Habitat Use and Adaptation

According to reference [16], the numerous rock dimensions and shapes were favorable habitats because they balanced the speed of water. This assumption is the same as in the habitat of *M. pauliani*, where their presence and abundance correlate with the rock presence and also the water depth, degree of slope, water velocity, and stream flow. 

Although forest degradation is a threat to the biodiversity of Ankaratra, this study showed that both savannah and forest represent important environments for this species to ensure its viability, a result similar to those previously put forward by the authors of reference [5]. Moreover, while the threats may result in greater water flow and floods, they are likely damaging tadpoles or washing them away in both savannah and forest, which represents a risk for their survival. However, the impact of forest degradation requires further analysis. So far, it has not had an immediate impact on the population of this species, as suggested by similar studies [31], but the long-term effects need to be analyzed. The Ankaratra Massif is vulnerable to various human activities that can have a negative impact on the survival of the species. According to the authors of reference [1], high altitudes are refuges for several species whose distribution is restricted and which are also more vulnerable. Fragmentation is one of the main threats to the disappearance of amphibians [32,33]. Logging, erosion, and livestock grazing leading to the trampling of watercourses remain the most significant threats. These threats lead to the destruction and pollution of watercourses, as in the case of Manotongana at 2167 m a.s.l., where the streams are very degraded and polluted. As a result, only one adult individual was recorded at this site in 2018, and none were recorded for three years thereafter.

We recommend strengthening monitoring through habitat patrols by forest rangers, especially in the savannah habitat which may be important for breeding. The establishment of a firebreak system around the forest is also essential to minimize the damage caused by bushfires. Environmental education, awareness-raising, and consciousness-raising within the village community must also be undertaken. This will allow them to understand the importance of the forest and make them aware that biodiversity conservation will benefit them in the long term. Finally, the livelihoods of the local population should be ameliorated through the provision of water from Ankaratra.

## 5. Conclusions

Overall, this study highlighted the vulnerability of *M. pauliani* to habitat degradation and provided an assessment of the population status. The species is found at altitudes between 1900 and 2378 m, and adults and juveniles were more abundant in forests than in savannahs, while tadpoles preferred the savannah habitat. Our results also showed how the number of individuals encountered in surveys varies across seasons and years. During the study, we noticed a decrease in the number of individuals encountered for each age class at the most degraded site, Manotongana. The viability of *M. pauliani* depends on the quality and quantity of its habitat on the Massif, specifically the stream, and its degradation has a direct impact on the survival of the population. To effectively manage and conserve *M. pauliani*, information is needed about their habitat preference, population number, distribution across altitudes, and the threats they face. Our study provides some of this critical baseline data. We hope decision-makers can use our results as a tool in the development of a conservation strategy with proactive measures to prevent the irreversible loss of one of the emblematic species of the Ankaratra Massif.

## Figures and Tables

**Figure 1 animals-13-02706-f001:**
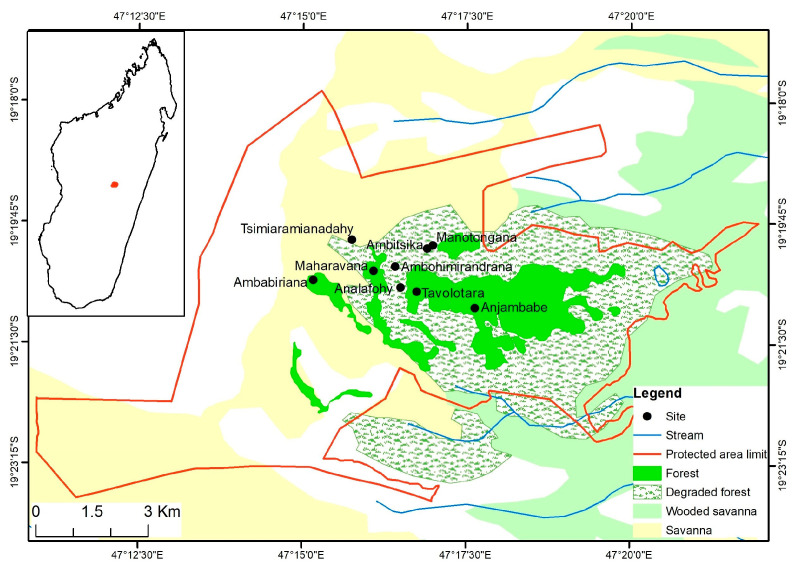
Map of Ankaratra showing the nine sampling sites.

**Figure 2 animals-13-02706-f002:**
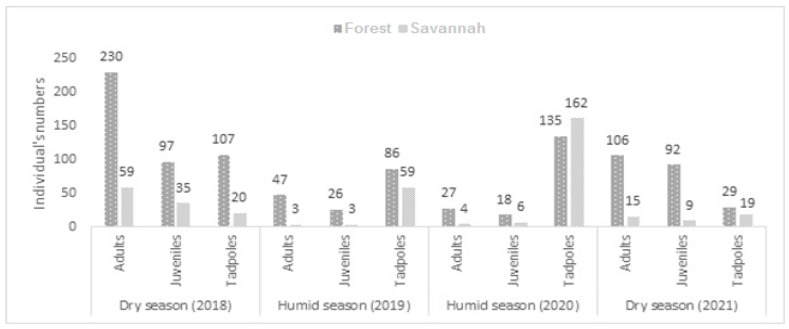
Numbers of individuals per age class counted according to habitat type for each season of fieldwork.

**Figure 3 animals-13-02706-f003:**
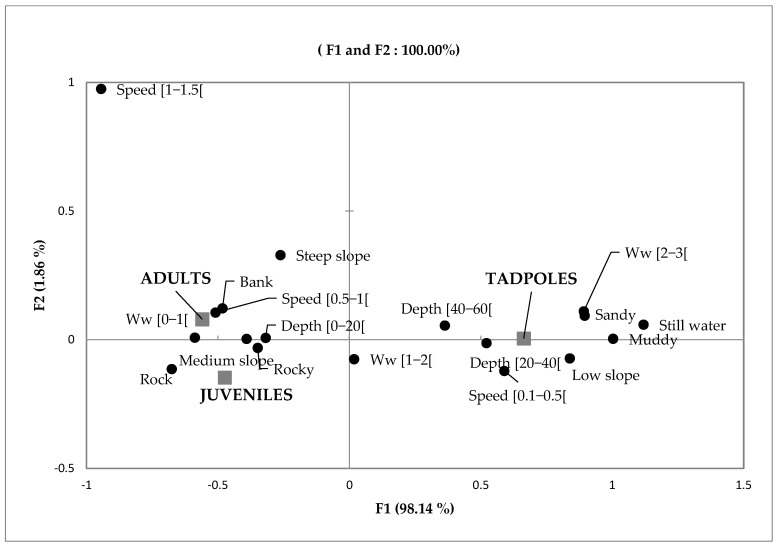
Projection of variables and individuals by developmental stage on the factorial plane (F1 and F2). Ww: water width.

**Figure 4 animals-13-02706-f004:**
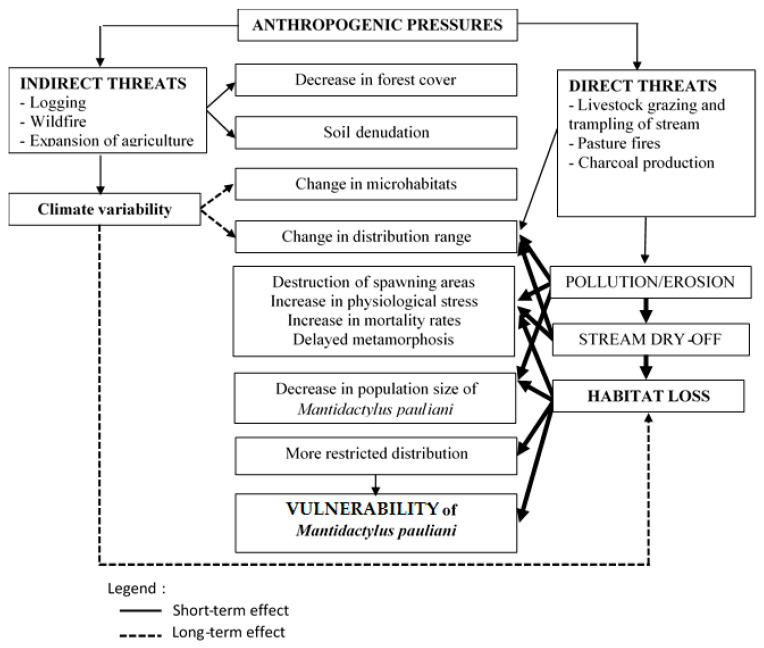
Diagram showing the relationship between habitat loss and the vulnerability of *M. pauliani*.

**Table 1 animals-13-02706-t001:** Description of the sampling sites.

Transect	Elevation (m)	Description
Anjambabe	1900	Natural forest
Tavolotara	1993	Natural forest
Maharavana	2100	Natural forest
Analafohy	2126	Degraded forest
Manotongana	2167	Edge ^1^
Ambaniriana	2166	Galery forest
Ambitsika	2202	Savanna
Ambohimirandrana	2244	Savanna
Tsimiaramianadahy	2378	Savanna

^1^ River separating a deforested area and bare soil.

**Table 2 animals-13-02706-t002:** Summary table of the numbers of *Mantidactylus pauliani* individuals counted by age class according to the altitude during each study season. A = Adult; J = Juveniles; T = Tadpoles; Alt = Altitude.

Alt	Dry Season (2018)			Total	Humid Season (2019)			Total	Humid Season (2020)			Total	Dry Season (2021)			Total
	A	J	T		A	J	T		A	J	T		A	J	T	
1900 m	36	16	24	76	2	7	34	43	0	0	12	12	3	0	12	15
1993 m	155	40	50	245	28	13	15	56	10	3	11	24	31	11	5	47
2100 m	-	-	-	-	-	-	-	-	9	8	48	65	26	14	0	40
2126 m	-	-	-	-	-	-	-	-	5	5	30	40	31	28	8	67
2166 m	38	39	11	88	17	6	13	36	3	2	34	39	15	39	4	58
2167 m	1	2	22	25	0	0	26	26	0	0	0	0	0	0	0	0
2202 m	36	29	9	74	2	2	44	48	2	5	61	68	11	8	5	24
2244 m	-	-	-	-	-	-	-	-	0	1	69	70	1	0	0	1
2378 m	23	6	11	40	1	1	15	17	2	0	32	34	3	1	14	18
Total	289	132	127	548	50	29	147	226	31	24	297	352	121	101	48	270

**Table 3 animals-13-02706-t003:** Number of *M. pauliani* males and females according to habitat type and season.

Season	Forest		Savannah	
	Female	Male	Female	Male
Dry season (2018)	87	158	24	20
Humid season (2019)	13	34	2	1
Humid season (2020)	6	21	3	1
Dry season (2021)	35	71	6	9

**Table 4 animals-13-02706-t004:** Evaluation of pressure.

Environment Types	Sites	Threat Types	Score			Evaluation
			Length	Intensity	Importance	
Forest environment	Anjambabe (1900 m)	Bushfires	1	1	1	Minor (3)
		Charcoal production	1	2	1	Minor (4)
		Logging	2	1	1	Minor (4)
		Livestock grazing	1	3	1	Medium (5)
	Evaluation total [Index of pressure PI]					16 [0.44]
	Tavolotara (1993 m)	Bushfires	2	1	1	Minor (4)
		Charcoal production	2	1	1	Minor (4)
		Logging	1	2	1	Minor (4)
		Livestock grazing	1	2	1	Minor (4)
	Evaluation total [Index of pressure PI]					16 [0.44]
	Maharavana (2100 m)	Bushfires	1	2	1	Minor (4)
		Charcoal production	1	2	1	Minor (4)
		Logging	1	1	1	Minor (3)
	Evaluation total [Index of pressure PI]					11 [0.41]
	Analafohy (2126 m)	Charcoal production	2	2	1	Medium (5)
		Logging	2	2	1	Medium (5)
		Livestock grazing	1	2	1	Minor (4)
		Expansion of agricultural land	1	1	1	Minor (3)
	Evaluation total [Index of pressure PI]					17 [0.47]
	Ambaniriana (2166 m)	Charcoal production	1	2	1	Minor (4)
		Logging	1	1	1	Minor (3)
		Livestock grazing	2	2	1	Medium (5)
	Evaluation total [Index of pressure PI]					12 [0.44]
Edge	Manotongana (2167 m)	Bushfires	2	3	1	Medium (6)
		Charcoal production	2	3	1	Medium (6)
		Logging	2	2	1	Medium (5)
	Evaluation total [Index of pressure PI]					17 [0.63]
Savannah	Ambitsika (2202 m)	Bushfires	1	2	1	Minor (4)
		Charcoal production	2	1	1	Minor (4)
		Feux de pâturage	1	2	1	Minor (4)
		Livestock grazing	2	3	1	Medium (6)
	Evaluation total [Index of pressure PI]					18 [0.5]
	Ambohimirandrana (2244 m)	Charcoal production	1	2	1	Minor (4)
		Livestock grazing	2	3	1	Medium (6)
	Evaluation total [Index of pressure PI]					10 [0.55]
	Tsimiaramianadahy (2378 m)	Bushfires	2	3	1	Medium (6)
		Livestock grazing	2	3	1	Medium (6)
		Charcoal production	1	2	1	Minor (4)
	Evaluation total [Index of pressure PI]					16 [0.59]

## Data Availability

The data presented in this study are available on request from the corresponding author. The data are not publicly available due to privacy or ethical restrictions.

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
