# Peer review of "Population Status and Vulnerability of Mantidactylus pauliani from Ankaratra Protected Area, Madagascar"

_animals, 2023, doi:10.3390/ani13172706_

Round 1

Reviewer 1 Report

I reviewed a manuscript by Oninjatovo Radonirina et al. about Mantidactylus pauliani. The authors conducted important ecological field work on a threatened species over four field seasons. Such studies are missing for most frog species in Madagascar, so the data they collected is very important to share. However, the conclusions made in the manuscript as currently written are not supported by the data. I provide 19 comments on the attached PDF directly. Here are my main recommendations:

1)    Re-frame the aim, objectives, and scope of the study throughout the manuscript. Present the study as evaluating habitat preferences, threats, and population status. Determining the effect of landscape change on the population is beyond the scope of the study and would require more precise estimates of abundance and measures of how the habitat has changed over a longer period of time.

2)      Consider other terms than “species distribution”, “ecological niche”, etc. (see attached PDF comments) which have specific meanings that differ from your results. For example, while raw counts is a measure of abundance, readers expecting “abundance” may seek population size estimates, and “species distribution” typically refers to geographic distribution rather than altitudinal range. Check/edit terminology throughout.

3)      Remove the note about color change and minimize the sex ratio results. See comments in PDF.

4)      Combine figures 3 and 4 into a single figure with stacked bars for easier interpretation (see comments in PDF). Make sure all figures have labelled axes and understandable legends.

5)      The writing is well-organized and flows well, but there are errors related to word choice, sentence structure, and grammar. I suggest running the manuscript through Grammarly (www.grammarly.com) or another tool and editing heavily. A few other writing suggestions:

-          Tense. Some of the methods and results are in present tense but should be in past tense. For example, “they remain stable” and “adults are more abundant” should be “they remained stable” and “adults were more abundant”

-          Active versus passive voice. Although this is a stylistic preference, I think the manuscript will be easier to read if sentences that are in the passive voice are changed to the active voice. For example, “A 100-meter transect was established with varying widths…” becomes “We established a 100-meter transect with varying widths…” See https://owl.purdue.edu/owl/general_writing/academic_writing/active_and_passive_voice/active_versus_passive_voice.html

-          Define all acronyms before use (for example, NPA in Figure 1 should be New Protected Area and PA in the manuscript title should be Protected Area

Overall, the study design does not have major flaws that prevent it from being published, as long as the results are not exaggerated or overinterpreted. Please see the PDF with my comments for more detailed suggestions to get this paper where it needs to be for publication.

Most of the errors I noticed related to word choice, sentence structure, and grammar. I was able to understand most of what you wrote, but often it was difficult and took some effort to piece together. Use Grammarly or other software to catch the grammatical errors. Word choice might be more difficult to correct, so consider showing the manuscript to other colleagues for help. The good news is that most of the writing in the manuscript is organized well, which is more important than the small things like grammar.

Author Response

Dear Reviewer, 

Thank you for your valuable time in peer-reviewing our manuscript. Please see the attachment as a reply to your comments. 

Reviewer 2 Report

The article presents interesting and significant information both of relevance to the conservation of an endangered amphibian, but also of general ecological interest.

There are numerous sections where the article is difficult to understand due to poor grammar and expression. The article should be carefully rewritten and presented to several others for critique before resubmission.

Author Response

Dear Reviewer, 

We thank you for reviewing our manuscript. We have made some important changes related to the English and the writing style as per your recommendations. Please see the attachment. 

Round 2

Reviewer 1 Report

The edits and changes you made greatly improved the manuscript. You now present three important results (distribution and elevational range, habitat use, and threats) and your results are supported by your data and analyses. Also, the updates to Figures 1 and 3 help with interpretation. I also appreciate your detailed methods related to threat pressure. Overall, very nice work and big improvements.

However, the manuscript is not ready for publication. The first minor area that needs work relates to the study objectives. I appreciate your effort in revising the last paragraph of the introduction (lines 63-70), but the section still overstates/misinterprets your study. Why use the terms “habitat fragmentation” and “population viability” when you did not study these? Instead, you studied the altitudinal range, habitat use, and threats. Align the scope of your paper and research aims with your methods and results.

The second and more important area that needs work is writing. The manuscript is difficult to understand because of many technical/grammatical errors. More importantly, there is a problem with word choice, where terminology is used incorrectly (please see my comments on the PDF). One option would be to run each sentence through a grammar checker or AI chatbot to make sure each section is easy to read. Alternatively, or additionally, a copy editor could work on this manuscript to fix it. Once the technical, grammatical, and word choice errors are corrected, the manuscript will be in much better shape.

For examples of the type of errors that need to be corrected, I proofread the Study Summary. Here are some of the errors in the first paragraph of your manuscript:

·         Line 13: “high” should be “highly”

·         Line 14: “dis-tribution” should be “distribution”

·         Line 15: “human activities caused a landscape changes that modify its…” should be “human activities have caused landscape changes that modified its…”

·         Line 19: “M. pauliani” should be italicized

·         Line 19: what is meant by “its abundance is observed”? the species occurs between 1900 m to 2378 m and is abundant between 1993 and 2166 m, perhaps?

·         Line 21: “width of the stream” should be “stream width"

·         Line 21: “which implies their vulnerability in the face of the degradation of these parameters” should be “which implies they are vulnerable to the degradation of these parameters”

·         Line 22: “oc-curence” should be “occurrence”

·         Line 23: “M. pauliani” should be in italics

Other common grammatical errors that make the article difficult to understand:

·         “in the mountain” versus “on the mountain”

·         Typos like line 133 “Spaciale” versus “Spatial”

·         Italicize scientific names

·         Errors related to articles like “the” and “a”

·         Montane versus mountain

·         Abbreviate genus after first use

·         Incorrect tense: for example, line 66 “this study will explore the…” should be “this study explores the…”

·         Duplicated sentences (for example, see line 218)

Writing in the passive voice instead of the active voice

Please also see my 7 comments directly on the PDF.

Run the manuscript through Grammarly or other service (for example, maybe https://quillbot.com/grammar-check) and/or have it copy-edited. There are too many errors for publication currently.

Author Response

Dear Reviewer,

We thank you for taking the time to review the manuscript and for your valuable suggestions for improvement. We have therefore made the necessary corrections and highlighted them to the manuscript. 

Please see attachement. 

Best regards, 

ONINJATOVO RADONIRINA Herizo

Reviewer 2 Report

The article makes a good contribution to species conservation and ecology. Please give it one last check through as you tidy up as per the PDF.

I have attached a PDF with a few grammatical changes for good English, and to make things clear. Please modify the article and represent to journal as editors require.

Author Response

Dear Reviewer,

We thank you again for your comments and suggestions. We have made the necessary corrections as per indicated in the manuscript. 

Kind regards, 

ONINJATOVO RADONIRINA Herizo.

Round 3

Reviewer 1 Report

Please see the attached Word document with the line-by-line edits that I suggest. All edits relate to writing, especially word choice and sentence structure. 

The quality of the English in the introduction is not too bad, but the summary, abstract, results, and conclusion all need substantial work. Please see my suggestions in the Word document that I attached.

Author Response

Dear Reviewer, 

We thank you again for the revision you have made to improve the quality of english in the manuscript. 

Please find attached the replies to your comments.

Sincerely yours, 

ONINJATOVO RADONIRINA Herizo
